# Measuring the relationship between museum attributes and visitors: An application of topic model on museum online reviews

**Hong Huo[1], Keqin Shen[1]\*, Chunjia Han[2], Mu Yang[2]**

**1** School of Management, Harbin University of Commerce, Harbin, China, **2** Department of Management, Birkbeck, University of London, London, United Kingdom

\* 1979320391@qq.com

**Data Availability Statement:** This paper used the Tripadvisor 4-core dataset which is publicly available on https://www.tripadvisor.cn/. And all

## Abstract

In recent years, cultural tourism has increasingly embraced museum visits. Museums serve as both cultural heritage guardians and integral parts of tourist destinations, significantly impacting visitor satisfaction and experience. Moreover, online museum reviews have become a crucial indicator of museum service quality, visitor experience, and public feedback in the digital age. An analysis of online reviews on major tourism websites and social media platforms can assist museums in developing appropriate management strategies. This study employed the structural topic model (STM) to analyze online museum reviews, identifying three primary attributes of museums and visitors' personal experiences, as well as 19 sub-attributes. The study confirmed that core offerings have a positive impact on visitor experience and satisfaction, while peripheral services and overall ambiance are also positively related to visitor experience and satisfaction. Furthermore, the results of structural equation modeling demonstrated that visitors' personal experiences have a positive impact on satisfaction. The results of structural equation modeling analysis support all seven hypothesized relationships. These findings will assist museum managers in developing effective management strategies and future plans.

## 1. Introduction

Museum tourism is a crucial component of cultural tourism and provides visitors with an opportunity to learn about the cultural and historical aspects of a city [1]. Museums contribute to society by providing education, entertainment, and recreation [2, 3]. In 2022, the International Council of Museums redefined museums as "a non-profit, permanent institution at the service of society that explores, collects, conserves, interprets, and exhibits tangible and intangible heritage [4]". However, with the development of society and the economy, the increase in the number of visitors to object museums has affected the operation of museums, resulting in negative phenomena such as long queues and loud noise, which bring a bad experience to tourists visiting museums [5]. Meanwhile, other cultural tourism products are becoming more interactive and engaging, making the static displays and explanations of traditional museums

relevant data are within the paper and its Supporting Information files.

**Funding:** Thanks to the Natural Science Foundation of Heilongjiang Province, No. LH2022G014 for the financial support of this paper.

**Competing interests:** The authors have declared that no competing interests exist.

appear relatively uniform and unappealing [6]. To address the challenges posed by increasing tourist numbers and competition, museums must adapt their management strategies. A comprehensive understanding of tourists' perspectives and perceptions of museum services, and enhance the effectiveness of their services to maintain their unique position and lasting appeal in cultural tourism markets.

Currently, there are many scholars who have studied the experience and satisfaction of museum visitors through questionnaires and interviews, as well as to understand visitors' motivations and intentions towards the museum, these studies use numerical data [7, 8]. However, due to the rapid development of digitization and informatization, an increasing number of museum visitors are opting to share their experiences on social media to express their satisfaction or dissatisfaction with the museum services. The online reviews generated by these visitors are also a valuable source of data for measuring visitor experience and satisfaction [9]. Online reviews have been suggested as a means of communicating cultural values in cultural tourism [10]. In the context of museum research, studies analysing the quality of museum services using online reviews are predominantly manual, with a few utilising automated analysis. Due to the large volume of online reviews, manual analysis can be time-consuming and labour-intensive, and it may introduce subjective evaluations, reducing the accuracy of the text analysis, automated analysis is more objective and reliable, and can save time [11]. Some scholars have suggested that analysing museum service quality first requires an understanding of museum attributes [5]. Zanibellato et al. used automated analytics to identify museum attributes (core offerings, peripheral services, ambiance) in online reviews and to assess how these attributes affect electronic word-of-mouth [12]. Orea-Giner et al. identified museum attributes through text mining and text analysis and added new attributes based on other researches [13]. Agostino et al. used Latent Dirichlet Allocation (LDA) to extract topics related to the quality of museum services, and identified three primary categories, such as museum cultural heritage, personal experience and museum services [14]. Attributes were identified by analyzing online reviews of visitors on social media, which is an accepted method [9, 15].

Although studies have been conducted on museum attributes, Orea-Giner et al. highlighted limitations in the research methods used to extract these attributes, therefore, there is a need to employ more effective methods for extracting museum service attributes [13]. In addition, Orea-Giner et al. noted that while there have been studies identifying museum attributes, no scholars have explored the relationship between museum attributes and visitor experience or visitor satisfaction [13].

To address these research gaps, this study employs structural topic model to extract comprehensive and scientific museum attributes from online reviews. The method is based on topic probability and is suitable for analysing user-generated content, such as online reviews and social media content [16]. The method is objective and accurate, surpassing manual analysis and other automated methods, it also allows for the inclusion of additional influences as covariates [17]. Then, the manuscript categorised the extracted topics using existing literature and factor analysis. The impact of museum attributes on visitors' personal experience and satisfaction, as well as the impact of visitors' personal experience on satisfaction, were analysed using exploratory factor analysis (EFA) and structural equation modeling.

In summary, this paper aims to answer the following research questions: How does the structural topic model approach aid in identifying museum attributes? How do these attributes impact the experience and satisfaction of museum visitors?

The remainder of the paper is structured as follows: The second part reviews the literature on museum attributes, visitor experience, and satisfaction, highlighting the relationships between them. The third section provides information on the content of online reviews, as well as data descriptions used in the structural topic model and the steps involved. There is

also factor analysis of extracted topics and analysis of unstructured data. In the fourth section, we present a discussion of the findings by incorporating the identified museum attributes as well as individual visitor experiences and satisfaction into the hypothesis model for analysis. The last remaining section provides an overview of the theoretical practical implications of the study research limitations and future research directions.

## 2. Literature review

### 2.1 Visitor experience and museum attributes

Visiting museums is a cultural consumer behaviour that aims to provide visitors with a memorable experience. Each experience is unique and creates lasting memories, which may influence visitors' future behaviours, such as generating a willingness to recommend and revisit [18, 19]. Chan argued that museums provide tangible (quality and quantity of services) and intangible (visitor perceptions and feelings) experiences for visitors, and are experiential places of consumption for entertainment, relaxation, and learning [20]. Tsao and Hsieh reported that experiences in the service environment are the embodiment of consumer feelings, senses and experiences [21]. According to Pine et al. experience is considered an economic product derived from goods and services [22]. Therefore, Schmitt proposed a new marketing approach, namely experience marketing, which is mainly aimed at transforming the consumer's consumption process into a holistic experience that results in a pleasurable experience [23]. Because museum visitors are independent individuals, they need different offerings and services and thus have different experiences [24, 25]. And found that various forms of "experiential marketing". impact consumer emotions and satisfaction. Therefore, it is essential to analyze visitor experiences. Museums should tailor their experiences to the needs of different visitors to develop appropriate demand strategies and improve the decisions of museum managers [26].

Increasingly, museum managers are realizing that the museum experience has an impact on visitors beyond collections and exhibitions. The visitor experience is a composite of images in the visitor's mind that is shaped by ambiance and services provided [27]. Falk divided the visitor experience into three phases: before, during, and after the visit. Before the visit, visitors check social media for basic museum information (opening hours, ticket prices, address, etc.) as well as visitor reviews [25]. Once inside, visitors experience the tangible and intangible services and products offered by the museum. These experiences continue after visitors leave and continue to share the experience on social media [28]. Falk described visitors' experiences in museums in terms of three different environments: socio-cultural environment, personal environment, and physical environment [25]. The term socio-cultural environment refers to the cultural background of the visitor, while the personal environment pertains to differences in their motivations, interests, and attitudes towards visiting the museum. The physical environment encompasses the museum's architecture and surrounding facilities. The analysis of museum attributes is closely related to the visitor's decision to visit or not. It is not only related to the visitor's personal cultural background but also to the different attributes of the museum, such as its exhibitions, collections, and physical services. Improving the quality of collections and adding exhibitions can enhance the visitor experience [29]. However, some museums believe that whether or not a visitor visits a museum is related to the visitor's level of education and standard of living [30]. Antón et al. and Muzellec et al. suggested that the use of electronic word of mouth to identify museum attributes is a current research hotspot [29–31]. This is a qualitative research method that focuses on reviews from Tripadvisor and various tourism websites as a way to identify destination and resource attributes [15]. Zanibellato et al. proposed three key museum attributes through textual analysis of online museum reviews: core

offerings, peripheral services and ambiance [12]. Agostino et al. extracted 14 museum sub-attributions through topic model (LDA) and categorized these topics into three key attributes: museum cultural heritage, personal experience and museum services [14]. These categories were defined based on visitor satisfaction research and online reviews of museum visitors in a similar context [12].

## 2.2 Visitor satisfaction

Visitor satisfaction is a crucial consideration for the tourism industry, particularly in the realm of tourism marketing. Satisfaction can be defined as the cognitive and affective evaluation of a product or service by the customer. De Rojas and Camarero defined satisfaction as the feelings generated by cognitive and affective aspects of goods and services, and the cumulative evaluation of components and features in the heel [32]. Thus, in the context of museum studies, satisfaction refers to the cognitive and affective responses generated by museum visitors. Meeting customer expectations indicates that they have received quality service and a high level of satisfaction. Therefore, museums must proactively understand and manage patrons' needs to meet their expectations [33]. According to Chin, satisfaction is not only used to assess customer expectations, but also to assess customer emotions and self-identity [34]. Chiappa et al. confirmed that customer satisfaction with museums is related to their emotions when visiting museums [35]. Falk argued that the high quality of service and museum experience provided by museums and the interaction with visitors can contribute to customer satisfaction [25].

Kempiak et al. investigated that the information conveyed by the museum, the ambiance, and the communication and engagement with visitors affect the visitor experience and thus satisfaction [36]. The product/service, visitor experience, and visitor satisfaction can be viewed as inseparable wholes, as good product or service leads to good experience and high satisfaction [37]. A positive correlation exists between visitor satisfaction and the number of positive online reviews generated [29]. When visitors are satisfied with their visit to a museum, they tend to re-visit, post positive reviews and be loyal to the museum [38]. When generating consumption behavior at a museum, customer satisfaction is reflected in the intention to recommend it to others [29–36]. The decision of visitors to share their experience on social media after visiting an attraction depends on their level of satisfaction. The higher the satisfaction level, the more likely they are to share a review [29]. Visitors also decide whether to recommend an attraction based on these positive or negative reviews.

## 2.3 Visitor experience, museum attributes, and visitor satisfaction: Hypothese development

From a tourism marketing perspective, cultural institutions offer a wide range of services. They not only provide core services, such as cultural products, but also peripheral services and create an ambiance for visitors [39]. The core of a museum refers to its collections, exhibitions, artworks, and other objects that visitors consume. These core offerings are crucial for the museum's sustainability. Peripheral services, such as audio commentary, lounges, and restrooms, are additional attributes that enhance the value of the core offerings. The visitor experience in a museum environment is greatly influenced by its ambiance, which includes factors such as visiting hours, patronage, and other conditions.

Attribute analysis can assist in identifying visitor experience, satisfaction, and behavioural intentions [40]. In the museum context, visitor satisfaction and experience can be analyzed to better understand museum performance [41, 42]. Empirical studies of museums' core offerings have shown that the content of museum exhibitions directly affects visitors' experience and satisfaction. Lin argued that a museum's core offerings include not only the "product"

offered by the museum, but also the added value generated by a visit to the museum, and also suggested that all of a museum's core offerings affect visitor satisfaction and experience, and ranks these core attributes when analysing the museum visitor experience [43]. In general, museums' core offerings have an educational value that not only creates a favourable learning environment for visitors, but also increases visitor satisfaction [44]. Currently, digitisation technology is being used as an important part of the museum's core offering, helping to improve visitor experience and satisfaction [45]. Zanibellato et al. study also showed that the most frequently cited positive attribute in the museum's online reviews was its core offerings [12]. Consequently, this study intended to verify the following hypothesis:

H1: The museum core offerings have a positive impact on the personal experience.

H2: The museum core offerings have a positive impact on visitors satisfaction.

In addition to the core exhibition, the museum's peripheral services are another important attribute that is an indispensable part of the overall visitor experience. Good peripheral services can eliminate inconveniences during the visit, enhance the visitor's immersive experience, and promote the visitor's sense of identity and belonging to the museum, which can effectively increase visitor satisfaction [12]. In recent years, museum practitioners have been actively engaged in creating a comfortable/healthy peripheral services for visitors, as this has a direct impact on the visitor experience [46]. In museums, peripheral service factors such as museum location, food and beverage facilities, and ticket reservations respond to visitors in terms of emotion/cognition [47]. Heesup et al. argued that museum peripheral services help visitors to effectively derive intellectual value and satisfactorily evaluate their overall museum experience [48]. As a result, we proposed the following hypothesis:

H3: The museum peripheral services have a positive impact on the personal experience.

H4: The museum peripheral services have a positive impact on visitors satisfaction.

In addition to core offerings and peripheral services, museum ambiance plays an important role in enhancing museum visitation. Museum ambiance is an environment designed to create stimuli that influence the perceptual experience and affective behaviour of visitors [49]. This environment includes factors such as temperature, odor, and lighting, and these environmental conditions are crucial for museum visitors [50]. Wu and Li, argued that these environmental conditions affect visitor experience and satisfaction [51]. Harrison and Shaw analyzed core experience, staff services, peripheral services and facilities, showing that staff services and ambiance had less impact on visitor satisfaction than visitor recommendations to others [52]. However, Huo and Miller concluded that staff service and ambiance had a significant effect on satisfaction [53]. Zanibellato et al. concluded that museum ambiance is an important driver of visitor satisfaction, while museum layout, temperature, and queuing are the key factors that influence the visitor's experience of visiting the museum [12]. Nurfatihah et al. found that the ambiance of a restaurant provides a pleasant experience for visitors [54]. It is believed that ambiance has an important mediating role in museum visitors and can enhance the attractiveness of museum [55]. In this regard, the following hypothesis were proposed in this study:

H5: The museum ambiance have a positive impact on the personal experience.

H6: The museum ambiance have a positive effect on visitors satisfaction.

Tourism literature considers museums as experiential consumption sites for learning, relaxation, and social interaction [20–37]. Chan argued that museums offer a "product" experience to visitors [20]. Pine et al. considered that museums provide a special value experience that

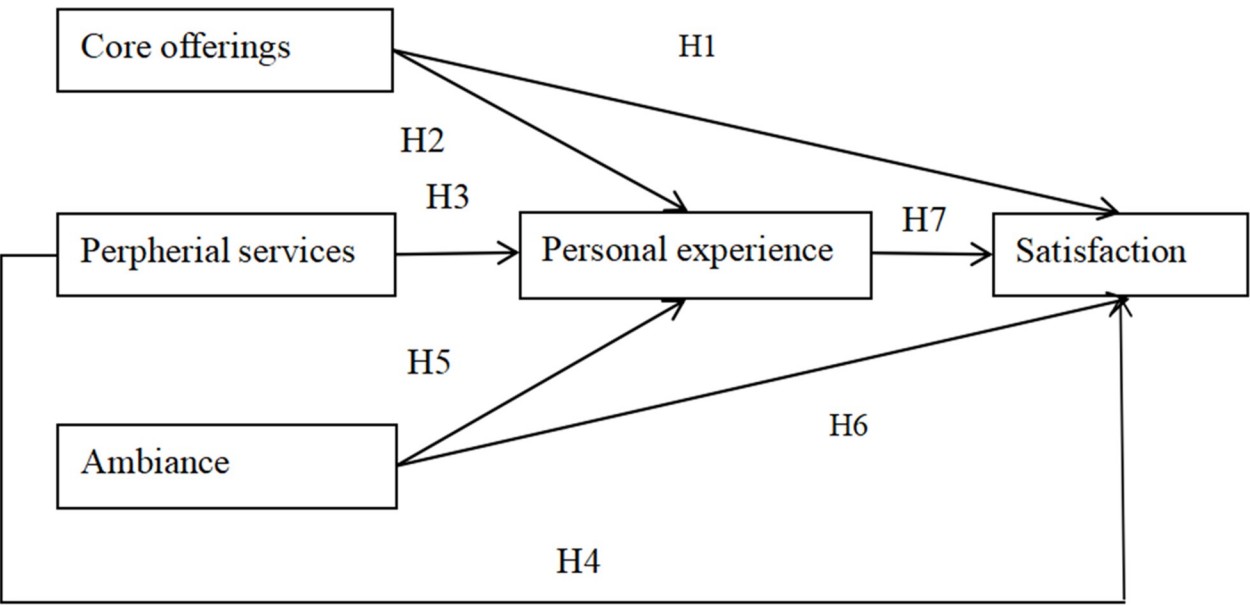

**Fig 1. Conceptual model and hypothesis.**

visitors are willing to pay [22]. Vu et al. suggested that there is a link between the visitor experience and satisfaction with the culturally relevant services provided by cultural institutions [56]. When the experience and behavioral intentions of tourists in the museum are understood, the attractiveness of tourist destinations to tourists will increase [57]. In previous studies of the museum experience, museum visitors have identified satisfaction as an important factor, and several key elements of the experience (compliments, entertainment, etc.) influence visitor satisfaction [58]. Therefore, this study proposed hypothesis 7.

H7: The visitor experience have a positive impact on satisfaction

The hypothesised conceptual model for this study is shown in Fig 1: specifically that the dimensions of museum attributes affect the personal experience and satisfaction of museum visitors, and that the personal experience plays a partially mediating role in the model.

## 3. Data and method

### 3.1 Data description and processing

The study's data was sourced from TripAdvisor, a prominent online travel site for tourism and hotel bookings that rapidly disseminates online review [59]. To account for the variability of reviews generated by different types of museums, we selected four distinct types: the British Museum, the National Gallery, the Natural History Museum, and the Victoria and Albert Museum. These museums cover a range of fields, including history, art, nature, and science. They are among the top ten most visited museums in the UK. To facilitate later data processing, we collected only English-language reviews, as English is the most commonly spoken language among tourists from various countries. The researchers used a crawler program to collect 118,966 English reviews from TripAdvisor between January 1, 2010, and April 1, 2023. For each review, we gathered essential information, including text comments, review times, and visitor ratings of the museum.

The text of each comment is considered as a document, then natural language techniques are applied to preprocess the document. The preprocessing process is based on previous

research [60]. The text underwent several pre-processing steps, including word tokenization, text normalization (converting uppercase letters to lowercase), elimination of numbers, white-space, and punctuation, and removal of stop words using lexical property (POS) tagging and the Python Natural Language Processing Toolkit (NLTK) [60]. The resulting corpus consisted of 102,728 documents, which were used for subsequent topic modeling analysis.

## 3.2 Extracting topics from structural topic model

This study uses the structural topic model (STM), a probabilistic approach to topic modeling. Each topic is made up of a distribution of words, and the words within each topic have a prob-ability of belonging to that topic [16]. This method was found to be suitable for analyzing user-generated content, such as social media content, online reviews, etc [17]. The nature and meaning of the corpus is understood through structural topic model that identify topics in documents and model relationships between these topics [61]. This method is considered superior to other topic modeling techniques as it is based on potential Dirichlet assignments. This allows for arbitrary information to be associated with the popularity of documents and topics through the use of covariates. These covariates have an impact on the popularity of top-ics, for example, comment ratings and review times can be associated with topics.

The flow of the STM for reviews text is shown in Fig 2 and the specific steps are as follows:

First, each topic of the document is extracted from a generalized linear model based on covariates and a set of logical normals. Such as Eq (1) which $d$ represents the document, $U_d$ represents the prevalence of each topic.

$$U_d \sim LogisticNormal(U_{d\gamma}, \sum) \tag{1}$$

Then, given a topic prevalence vector associated with a particular topic $Z_{d,n}$, the process is as follows, which $n$ is the index of each word in the document $d$.

$$Z_{d,n} \sim Multinomial(U_d) \tag{2}$$

Next, assign the words of each document to the topic. Where $\beta_{d,z}$ represents probability, it refers to the probability of choosing a word $W$ to fill position in the document $d$ given the

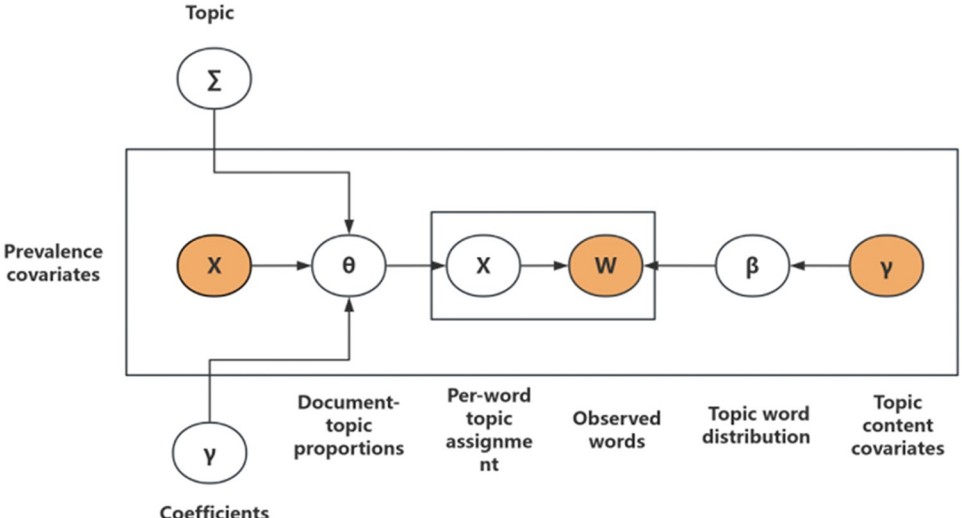

**Fig 2. A graphical illustration of STM using plate notation.**

subject variable.

$$W_{d,n} \sim Multinomial(\beta_{d,z}) \tag{3}$$

Then, we used the STM package in the R programming language to build our analytical model, the visitor reviews text as a document input, the popularity function is as follows:

$$Prevalence \sim rating + s(reviewtime) \tag{4}$$

where s is the smoothing function of time, and review time is one of the covariates of topic prevalence, indicating the time when the review was written.

Finally, we decide on the number of topics K which is an important user-specified parameter of STM and helps to achieve substantive interpretation of the outcomes of the modelling [62]. Using function search K from the stm and furrr packages in R, the number of topics was selected using three criteria [63]:

i. Heldout likelihood;

ii. Semantic Coherence of the words to each topic and;

iii. Exclusivity of topic words to the topic.

To determine the appropriate number of topics to generate, we began with an initial estimate of 5 topics and increased by increments of 2 until 30 topics were evaluated. We then selected the 19 topics that best met the three chosen criteria. The researchers assigned labels to the topics based on previous literature, using samples of the first 7 High Prob and FREX words. **Table 1** provides an estimated number of topics, as well as the assigned tags and 7 High Prob and FREX words.

### 3.3 Data analysis

This study utilizes SEM to test the research model, ensure model fit, and test the proposed hypothesis. SEM is valid for path analysis models with moderating and mediating variables [64]. SEM provides factor analysis and multiple regression analysis for the analysis of hypothesized relationships between observable and unobserved variables, a multivariate analysis technique that determines the relationship between variables consistent with the data sample [65, 66]. In this study, we extracted 19 observed variables from unstructured data using the STM package in R language to measure the hypothesis model presented above. We then categorized the measured variables based on topic analysis as common features of the topic distribution categories. The categories used in this study to describe museums are based on previous research. They are divided into three categories for the museum itself and the personal experiences of visitors. The 19 observed variables are grouped into four subcategories. The first subcategory measures the museum's core topics and the "product" it offers [12]. The second part utilises 9 variables to identify the services surrounding the museum. The third part comprises four variables intended to measure the museum's ambiance. The fourth category pertains to the personal experiences of museum visitors [14]. To measure visitor satisfaction, this study employs visitor review ratings on a five-point scale, ranging from very dissatisfied (1) to very satisfied (5) [67]. **Table 2** contains the operationalization of all variables, sources, and related studies.

Descriptive statistics and exploratory factor analysis (EFA) were conducted before further analysis to classify museum attributes in this study. During the extraction phase, principal component analysis was performed using orthogonal rotation. The eigenvalue criterion was used in this study to identify the relevant dimensions of museum attributes, this criterion

**Table 1. Topic summary: 102,728 reviews from 1 January 2010 and 1 April 2023.**

| Topic No. | Topic Label | Propotion | Top words |
|---|---|---|---|
| Topic 1 | Experience with kids | 5.7% | " Highest Prob: love, child, kid, year, great, old, famili "<br>" FREX: adult, kid, fun, young, daughter, old, child " |
| Topic 2 | Friendly staff | 3.7% | " Highest Prob: well, worth, help, staff, easi, friend, inform "<br>" FREX: help, well, laid, easi, organ, done, clean " |
| Topic 3 | London museums | 12.2% | " Highest Prob: museum, visit, thi, london, place, histori, time "<br>" FREX: histori, london, must, definit, trip, thi, visit " |
| Topic 4 | Entrance hall | 2.9% | " Highest Prob: build, entranc, main, hall, enter, floor, architectur "<br>" FREX: hall, entranc, main, ceil, grand, enter, structur " |
| Topic 5 | Queuing | 4.4% | " Highest Prob: dinosaur, queue, get, long, walk, wait, section "<br>" FREX: station, tube, zone, earthquak, earth, mammal, volcano " |
| Topic 6 | Collections | 3.5% | " Highest Prob: display, sculptur, design, fashion, object, room, modern "<br>" FREX: furnitur, ceram, jewelri, silver, cast, islam, iron " |
| Topic 7 | Cultural relics | 5.1% | " Highest Prob: british, museum, egyptian, stone, rosetta, artifact, ancient "<br>" FREX: rosetta, elgin, parthenon, marbl, stone, assyrian, mummi " |
| Topic 8 | Crowds | 6.4% | " Highest Prob: veri, lot, interest, good, crowd, busi, realli "<br>" FREX: lot, busi, veri, crowd, over, weekend, air " |
| Topic 9 | Visiting time | 12.9% | " Highest Prob: see, day, time, hour, much, spend, get "<br>" FREX: day, spend, hour, spent, everyth, plan, whole " |
| Topic 10 | Museum preference | 4.5% | " Highest Prob: museum, find, interest, found, item, area, differ "<br>" FREX: albert, find, victoria, found, might, may, pref er " |
| Topic 11 | Praises | 5.3% | " Highest Prob: one, collect, world, best, museum, thi, miss "<br>" FREX: best, world, class, ever, one, simpli, greatest " |
| Topic 12 | Location | 4.8% | " Highest Prob: art, galleri, paint, work, nation, free, mani "<br>" FREX: art, squar, trafalgar, lover, artwork, galleri, famou " |
| Topic 13 | Artist | 2.6% | " Highest Prob: paint, galleri, van, gogh, nation, monet, room "<br>" FREX: sainsburi, goya, constabl, turner, leonardo, rembrandt, titian " |
| Topic 14 | shopping | 4.7% | " Highest Prob: beauti, wonder, alway, shop, build, someth, gift "<br>" FREX: gift, alway,shop, varieti, wonder, beauti, wide " |
| Topic 15 | Guided Tours | 3.5% | " Highest Prob: tour, guid, audio, highlight, nan, use, inform "<br>" FREX: nan, audio, tour, guid, talk, listen, night " |
| Topic 16 | Booking and tickets | 5.2% | " Highest Prob: exhibit, special, book, ticket, see, went, time "<br>" FREX: ticket, advanc, book, exhibit, mcqueen, special, dior " |
| Topic 17 | Unfriendly staff | 5.1% | " Highest Prob: peopl, staff, get, read, one, disappoint, bag "<br>" FREX: poor, bag, push, rude, peopl, cloakroom, told " |
| Topic 18 | Café and dining | 5.4% | " Highest Prob: free, cafe, good, food, donat, great, restaur "<br>" FREX: food, tea, coff, cake, restaur, eat, lunch " |
| Topic 19 | Enjoyable experience | 2.1% | " Highest Prob: enjoy, room, time, even, around, look, way "<br>" FREX: thorough, enjoy, away, even, way, wife, feel " |

**Table 2. Operationalization and sources of all variables.**

| Variable | Operationalization | Source | Ralated studies |
|---|---|---|---|
| **Core offerings** | Percentage of topics related to core cofferings in each review | Text mining | [13]<br>[12] |
| **Peripheral services** | Percentage of topics related to peripheral services in each review | Text mining | [16]<br>[68] |
| **Ambiance** | Percentage of topics related to ambiance in each review | Text mining | [13]<br>[69] |
| **Personal experience** | Percentage of topics related to personal experiencein each review | Text mining | [14] |
| **Satisfaction** | Rating of reviews | TripAdvisor site | [67] |

recommends retaining only communal factors with eigenvalues greater than one [70]. In factor analysis, each underlying factor is associated with an eigenvalue derived from the covariance matrix. If the eigenvalue of a factor exceeds one, it indicates that the factor explains more variability than any individual raw variable, thus justifying its inclusion in the final factorial model [70]. When deciding whether to keep a variable, we also consider three other criteria: variables with non-conformance factor loads greater than 0.5, cross-conformities less than 0.5, and commonalities greater than 0.5 [71]. The objective of this study is to evaluate the impact of museum properties on visitor experience and satisfaction by building structural equation models in Amos 22.0. The study employed a two-stage inspection procedure. The first stage evaluated the measurement model, using Validating Factor Analysis (CFA) to assess convergence and discriminant validity of reliability and measurement variables [72]. The second stage is through path analysis to test the study hypothesis.

## 4. Results

### 4.1 Exploratory factor analysis

Pine et al. defined EFA as the use of extreme value rotation analysis to simplify the number of items and assess potential study dimensions. In this study, 19 museum measures were evaluated [22], the Kaiser-Meyer-Olkin (KMO) (0.945) and Bartlett's spherical test had a chi-square value ($\chi^2$ = 2077176.146). Due to the larger data set, the chi-square values were also larger. Therefore, the data are suitable for factor analysis. The study's results indicate that four factors accounted for 83.98% of the total variance. Factor 1, which contained three items, was identified as the core offerings and explained 14.62% of the total variance with a composite reliability value of 0.903. Factor 2, labelled as peripheral services, included nine items and explained 35.94% of the total variance with a composite reliability value of 0.957. Factor 3, defined as ambiance, included 4 items and accounted for 18.94% of the total variance with a composite reliability value of 0.897. Three items loaded onto factor 4 and were categorised as personal experience, explaining 14.48% of the total variance, with a combined reliability value of 0.939, as presented in the **Table 3**.

### 4.2 Measurement and structural model

To analyze the measurement model, four steps are required: measurement item reliability, structural reliability, convergent validity, and discriminant validity. The first step involves checking the Cronbach coefficients for individual items, which should all exceed a threshold of 0.7 [73], which indicates that the measurement items are feasible for the current research project [74]. Secondly, composite reliability values were calculated and found to be greater than the acceptable level of 0.7, indicating that the measurement items are internally consistent [34]. Thirdly, the study confirmed the existence of convergent validity by extracting mean variance values greater than 0.5. Finally, to achieve discriminant validity, it was necessary to ensure that the square root of the average variance extracted (AVE) for each construct was greater than the correlation between the constructs [22]. **Table 4** shows that the validated factor analysis models all achieved the desired value, indicating a good model fit. **Table 5** shows that each extraction met the requirements for convergent validity, with an average variance and CR value above 0.7 [75]. The square root of the AVE of each construct in the table is greater than the correlation between the constructs, indicating that the AVE of each construct is greater than the variance between it and the other constructs. Therefore, the data in this study satisfied the research criteria for discriminant validity.

**Table 3. Exploratory factor analysis.**

| items | Attributes | | | Personal experience |
|---|---|---|---|---|
| | Core offerings | peripheral services | Ambiance | |
| Collections | 0.901 | | | |
| Cultural relics | 0.826 | | | |
| Artist | 0.903 | | | |
| Friendly staff | | 0.912 | | |
| London museums | | 0.938 | | |
| Museum preference | | 0.883 | | |
| Shopping | | 0.795 | | |
| Guided Tours | | 0.940 | | |
| Booking and tickets | | 0.813 | | |
| Unfriendly staff | | 0.823 | | |
| Café and dining | | 0.862 | | |
| Location | | 0.936 | | |
| Entrance hall | | | 0.930 | |
| Queuing | | | 0.937 | |
| Crowds | | | 0.893 | |
| Visiting time | | | 0.824 | |
| Experience with kids | | | | 0.913 |
| Praises | | | | 0.920 |
| Enjoyable experience | | | | 0.950 |
| % of explained variance | 14.62% | 35.94% | 18.94% | 14.48% |
| Cronbach's α | 0.903 | 0.957 | 0.897 | 0.939 |

**Note(s):** Extraction method: principal component analysis Rotation method: varimax with Kaiser normalisation KMO = 0.945, Bartlett's test = 0.000.

## 4.3 Hypothesis testing

**Table 6** presents the results of the structural equation model used to test the research hypothesis. The analysis showed that the core offerings of the museum have a positive impact on visitors' personal experience and satisfaction (β = 0.109, P<0.000; β = 0.082, P<0.000), supporting H1 and H4. Additionally, the museum's peripheral services were found to have a positive impact on visitors' personal experience and satisfaction (β = 0.384, P<0.000; β = 0.228, P<0.000), supporting H2 and H5. The museum's ambiance have a positive impact on visitors' personal experience and satisfaction (β = 0.327, P<0.000; β = 0.095, P<0.000), supporting H3 and H6. Personal experience have a positive effect on visitors' satisfaction (β = 0.104, P<0.000), and supporting H7.

## 5. Discussion

The paper has two research objectives. Firstly, we aim to identify museum attributes from unstructured data using a structural topic model. This will establish a more effective method

**Table 4. Structural equation fit tables.**

| Model Fit Indicators | RMR | NFI | IFI | TLI | CFI | RMSEA |
|---|---|---|---|---|---|---|
| Desired value | ≤0.080 | ≥0.900 | ≥0.900 | ≥0.900 | ≥0.900 | ≤0.10 |
| Fit value | 0.001 | 0.932 | 0.932 | 0.920 | 0.932 | 0.097 |

**Table 5. Correlation matrix for latent constructs, AVE and composite reliability.**

|  | Core offerings | Peripheral services | Ambiance | Personal Experience |
|---|---|---|---|---|
| **Core offerings** | 0.877 | | | |
| **Peripheral services** | 0.561 | 0.880 | | |
| **Ambiance** | 0.485 | 0.544 | 0.897 | |
| **Personal Experience** | 0.483 | 0.623 | 0.589 | 0.928 |
| **CR** | 0.909 | 0.968 | 0.943 | 0.949 |
| **AVE** | 0.770 | 0.773 | 0.803 | 0.861 |

**Note(s):** AVE = Average Variance Explained and CR = Composite Reliability. All inter-correlation coefficients are significant at *p < 0.05 and

**p < 0.01. Italics Diagonal figures represent the square root of the AVE; sub-diagonal figures are the latent construct for inter-correlations.

for measuring museum attributes and evaluating the quality of museum services. Compared to the commonly used potential Dirichlet assignment method, STM can incorporate useful variables as covariates in the form of topic proportions and discourse word prior distributions. This text mining approach generates topics that are more relevant and have a stronger qualitative interpretation than traditional text analysis methods, such as dictionary scoring. Additionally, the study analysed the impact of the identified attributes on visitor satisfaction and experience. This study aims to help museum managers identify the key service attributes that impact visitor satisfaction and develop effective management strategies to improve the quality of museum services. We tested the research hypothesis by text mining online reviews of four UK museums to achieve the objectives of this paper.

The paper's findings highlight the influence of museum service attributes on visitors' personal experience and satisfaction. The path analysis coefficients indicate that a museum's core offerings have a positive impact on visitor experience and satisfaction. The museum's core offerings are its collections and artifacts, which are educational in nature and the primary reason for visitors to choose to visit. It is important to note that these offerings are the sub-attributes of the museum. However, this is not consistent with the findings of previous studies, which concluded that educational value is not an important factor for tourists to purchase museum cultural products, due to the low level of perceived quality, social and price value of museum cultural products [76]. Conti et al. demonstrated that museum core offerings have a positive impact on visitors' personal experience and satisfaction, and are a key factor in generating visitors' intention to recommend them [69]. Therefore, museums should focus on their core offerings to improve the quality of their services.

Museum peripheral services (shopping, dining, guided tour facilities, etc.) have less of a positive impact on the personal experience and satisfaction of museum visitors compared to core offerings. Zanibellato et al. noted that peripheral services of museums encompass multiple sub-attributes, for instance, ticket booking services offer convenience for tourists, additionally,

**Table 6. Hypothesis testing.**

| Path | Estimate | β | S.E. | C.R. | P | Supported? |
|---|---|---|---|---|---|---|
| Core offerings→Personal experience | 0.087 | 0.109 | 0.003 | 32.582 | *** | Yes |
| Peripheral services→Personal experience | 0.258 | 0.384 | 0.002 | 111.908 | *** | Yes |
| Ambiance→Personal experience | 0.336 | 0.327 | 0.003 | 100.890 | *** | Yes |
| Core offerings→Satisfaction | 0.870 | 0.082 | 0.043 | 20.392 | *** | Yes |
| Peripheral services→Satisfaction | 2.025 | 0.228 | 0.039 | 52.360 | *** | Yes |
| Ambiance→Satisfaction | 1.293 | 0.095 | 0.056 | 23.244 | *** | Yes |
| Personal experience→Satisfaction | 1.371 | 0.104 | 0.058 | 23.796 | *** | Yes |

some museums are located in busy areas, providing more accessible transportation for tourists [12]. However, Conti et al. reported that certain sub-attributes of peripheral service attributes, such as gift stores, ticket reservation services, food and beverage services, have a negative impact on visitor satisfaction. These sub-attributes can produce a large number of negative effects, leading to negative perceptions of the experience [69]. Meanwhile, Shao et al. found that visitors' dissatisfaction with museums is primarily due to poor tour guide and food and beverage services, as well as overpriced exhibits [77]. The mismanagement of museums is the main cause of these negative peripheral services. Therefore, museum managers must pay close attention to the management of peripheral services to avoid reducing visitor satisfaction and patronage due to mismanagement.

Ambiance is the third most important attribute, Forrest proved that museum ambiance has an impact on the visitor's personal experience and satisfaction, but did not determine whether it is positive or negative [46]. Kılıçarslan and Caber argued that in cultural heritage tourism, visitor perception is another determinant of visitor experience and satisfaction [78]. Therefore, museum ambiance has a significant impact on tourist satisfaction. The results of this study proved that museum ambiance have a positive impact on visitor experience and satisfaction. Zanibellato et al. identified the sub-attributes of ambiance, such as crowding, queuing, and visiting time, as key factors that affect visitor satisfaction [12]. Shao et al. pointed out in his study that the factor of crowding can have a significant negative impact on visitors' experience and satisfaction [77]. The sub-attributes below the ambiance are mainly caused by the museum's mismanagement. For example, the failure to control the number of visitors results in long queues and reduces the time available for tourists to visit the museum. Additionally, the lack of reasonable paths in the browsing area leads to overcrowding and other issues. Mismanagement can have a negative impact on visitors' personal experience and satisfaction. Therefore, museum managers should take measures to create a favourable museum ambiance for visitors to enhance their experience and satisfaction.

The museum co-creates the visitor's personal experience, and this paper validates the model of the impact of different museum attributes on visitor experience and satisfaction. The discussion above reveals the impact of each attribute on visitor experience and satisfaction. Additionally, visitors' personal experience also affects their satisfaction. Loureiro and Ferreira showed that visitors can directly or indirectly influence visitor satisfaction by visiting museums [79]. Vu et al. highlighted the correlation between visitor experience and museum services, they argue that the museum's visitor experience plays a crucial role in enhancing visitor satisfaction, particularly in relation to children's play and enjoyable experiences. It is important for museums to provide praise-worthy services to ensure a positive visitor experience [56]. Vesci et al. demonstrated that visitor personal experience has a significant impact on satisfaction. Therefore, these results are in line with previous research [80].

## 6. Implications

### 6.1 Theoretical implications

From a theoretical perspective, this paper makes three main contributions. Firstly, this study is the first to use the structural topic model technique to analyse TripAdvisor online reviews in the museum context. Specifically, the study integrates the review time and review rating of museum online reviews into the STM model as covariates. This approach aims to excavate and explain new dimensions and dynamic changes of the museum attributes that have not yet been sufficiently researched in the existing literature. Secondly, this study combines STM techniques with structural equation modeling (SEM) to address the gap in research methodology in the field of empirical museum research.

Secondly, although previous studies have recognized the importance of insights into visitor experience from User-Generated Content [10], there is a lack of social media-based museum service attributes [12]. Therefore, this paper not only enriches the research on museum attributes, but also introduces new service attribute indicators that can reflect the unique value of museums on the basis of existing research literature, which sets a new paradigm for future empirical research in related fields, and strongly promotes the innovation of museum management and marketing strategies in the digital era.

Thirdly, this study proposes the impact of museum attributes on visitors' personal experience and satisfaction, and the impact of visitors' personal experience on satisfaction by integrating the literature on museum marketing and museum attributes. Then, relevant factors are integrated into a comprehensive model to test hypothesis. It provides the basis for a conceptual framework that encompasses museum attributes and visitor experience and satisfaction. This study complements the hypothesis of the influence of museum attributes on tourists proposed in the study by previous research [13].

## 6.2 Practical implications

This study provides important insights for practitioners and managers by extracting museum attributes through online reviews. Based on this study, museum managers and marketers can adopt practical strategies and methods to improve the quality of museum services when enhancing visitor experience and satisfaction.

Firstly, identifying the core offerings of a museum can help to increase the positive experience and satisfaction of visitors. Museums should develop and use more innovative forms of "products" to enhance the personal experience of visitors. In terms of the content of innovative products, the value and significance of the museum as a World Heritage Site can be fully explored with visitors. For example, museums can hold diversified exhibitions, hold joint exhibitions in several museums, or open thematic exhibitions. Meanwhile, with the rapid development of science and technology, museums are gradually transforming into diversified, digitalised and interactive educational and entertainment venues. To better interpret the resources in the collection, attract audiences of different ages and provide a deeper visitor experience. Museums can also actively introduce a range of cutting-edge technologies. For example, the use of holographic projection to bring static cultural relics to life and vividly present them to the audience, enhancing the fun and educational value of the exhibition; or the use of artificial intelligence to help the museum provide intelligent tours, using speech recognition and natural language processing technology to provide personalised explanations to meet the needs of different audiences. AI can also be used to analyse visitor behaviour and interest preferences to optimise exhibition layout and content recommendations, creating a more intimate visitor experience.

Secondly, this study shows that museum peripheral services and ambiance have a significant positive impact on both visitor experience and satisfaction. However, peripheral services and ambiance are particularly vulnerable to negative impacts on visitors due to mismanagement. For example, overcrowding and long queues are common at peak times, which exacerbates visitor fatigue and diminishes the museum experience. Meanwhile, visitors perceive high admission prices, and in-museum catering and souvenir services do not meet their expectations in terms of quality, variety or value for money, which can lead to negative feedback. In addition, the extra cost of guided tours can leave visitors feeling dissatisfied, especially if they do not receive a high quality interpretation service. The service attitude of staff also has a direct impact on the image and reputation of the museum.

To improve visitor experience and satisfaction, museum managers should identify strategies to improve service quality on all fronts. For example, consider introducing a reservation system or flow restriction measures to avoid overcrowding in the museum, to ensure that visitors have enough space and time to look carefully at the exhibits, and to reduce queuing time to improve the smoothness of the tour. Formulate reasonable price concessions for holidays and introduce discount tickets or family packages to attract more potential visitors; improve catering facilities and pay attention to the cleanliness and elegance of the environment; and organise regular staff training courses to enhance the professionalism and service attitude of staff in guided tours, receptions and other service positions, so that tourists can feel the warmth and thoughtfulness of the service.

In summary, the results of this study highlight the relationship between museums and visitors, and the different influences that each museum attributes to visitors. Therefore, in order to achieve museum marketing goals, museum marketing plans should focus specifically on visitor experience and satisfaction and not be limited to "product" displays. They should also meet the needs of visitors in the increasingly competitive cultural tourism landscape.

## 7. Conclusions, limitations and future research directions

Despite the increase in research on museums in recent years, there is still a lack of empirical research on museum service attributes. This study seeks to understand museum service attributes by analysing visitor reviews of four UK museums on TripAdvisor and whether these attributes have a positive impact on individual visitor experience and satisfaction. The main findings of this study add new contributions to the field of research: (1) this paper uses structural topic model techniques to extract 19 topics from the reviews and introduces new sub-attributes based on existing research that identifies the main attributes of museums [12]; (2) museum core offerings positively affect visitors' personal experience and satisfaction; peripheral services are positively related to visitors' personal experience and satisfaction; (4) museum ambiance positively affects visitors' personal experience and satisfaction; (5) visitors' personal experience positively affects satisfaction. It is worth noting that personal experience can be considered as a mediating variable in the model, which has a significant mediating effect on the path of the influence of attributes on satisfaction (this is not the focus of this study and is not discussed in the results section). The findings of the study contribute to the development of museum theory and provide a basis for future research directions that can inform the development of marketing strategies for museum management.

It is worth noting that the sample reviews collected in this study were limited to four museums in the UK and were exclusively in English. It is possible that visitors from diverse cultural backgrounds may have different perceptions of museums. Therefore, it would be beneficial to consider variations in museum attributes across cultural backgrounds in future studies. Moreover, due to the limited literature on museum attribute research, it can be challenging to identify all museum attribute dimensions. It is uncertain whether a roundtable discussion method could be employed to determine attributes, which may result in more precise outcomes. However, we could explore alternative methods to ensure a comprehensive understanding of the subject.

Considering the research limitations mentioned above, this paper proposes future research directions. Firstly, this study only discusses the extraction of museum attributes from online reviews. Future research could conduct a more in-depth qualitative study by delving deeper into the emotions of online reviews to understand the visitor's experience of visiting the museum on an emotional level. Furthermore, the study did not analyse the impact of variables such as the type of visiting tourists and season on visitor experience and satisfaction. Finally,

the sample used in this study only included English reviews, and other languages were not analysed. Future research could expand to a more diverse sample of studies and compare the results of analysing these different samples.

## Supporting information

**S1 File.**
(ZIP)

## Author Contributions

**Conceptualization:** Chunjia Han.

**Methodology:** Keqin Shen.

**Supervision:** Hong Huo.

**Validation:** Mu Yang.

**Writing – original draft:** Keqin Shen.

**Writing – review & editing:** Keqin Shen.

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
