## [Decision Letter · Decision Letter 0]

26 Mar 2024

PONE-D-24-02778Measuring the relationship between museum attributes and visitors from unstructured data: A topic modeling application on museum visitors’ online reviewsPLOS ONE

Dear Dr. Shen,

Thank you for submitting your manuscript to PLOS ONE. After careful consideration, we feel that it has merit but does not fully meet PLOS ONE’s publication criteria as it currently stands. Therefore, we invite you to submit a revised version of the manuscript that addresses the points raised during the review process.

We have received the reports from our advisors on your manuscript and read them carefully. We think the reviewers provided very good assessments and recommendations. Based on the advice received, we feel that your manuscript could be reconsidered for publication should you be prepared to incorporate revisions.

We look forward to receiving your revised manuscript.

Kind regards,

Andrea Fronzetti Colladon, Ph.D.

Academic Editor

PLOS ONE

2. In your Methods section, please include additional information about your dataset and ensure that you have included a statement specifying whether the collection and analysis method complied with the terms and conditions for the source of the data.

Reviewers' comments:

Reviewer's Responses to Questions

**Comments to the Author**

1. Is the manuscript technically sound, and do the data support the conclusions?

Reviewer #1: Partly

Reviewer #2: Partly

2. Has the statistical analysis been performed appropriately and rigorously? 

Reviewer #1: I Don't Know

Reviewer #2: Yes

3. Have the authors made all data underlying the findings in their manuscript fully available?

Reviewer #1: Yes

Reviewer #2: Yes

4. Is the manuscript presented in an intelligible fashion and written in standard English?

Reviewer #1: Yes

Reviewer #2: Yes

5. Review Comments to the Author

Reviewer #1: The recently revised ICOM definition of museums should be provided, the one from 2007 is old.

"However, with socio-economic development, museums are increasingly threatened by cultural tourism" - please elaborate on that, it is not clear. The sentence that follows contradicts this "threat".

Hypothesis development - why H2, H3, H5 and H6 would have a negative impact? Yes, but only if they are poorly managed, otherwise they may greatly enhance both the experience and satisfaction. Try to re-phrase the hypotheses. I don't understand why you introduce another hypothesis (H7), besides there is only one hypothesis while you are referring to it in the plural. In general, the paper would benefit from a lower no. of hypotheses, try to reduce and re-phrase.

Some language editing is required (e.g. spaces between words and brackets; "Kempiak et al., (2017)shows" - show; "it to others[28-35]. whether" - capital W; "Hypothese" - hypothesis; "Lin, (2009) the study identifies" - ???; "core offerings has" - have; "peripheral services has" - have; "all belong to one of the top ten most visited museums" - do you mean "all belong to the top ten most visited museums"; "perpherial" - peripheral; etc.).

This is not a sentence: "Determine the appropriate number of factors (museum attribute categories), according to Kaiser, (1960) eigenvalue rules (keeping factors with eigenvalues greater than 1)[60]." - add the missing parts.

Methodology - "Cranbach" - it is "Cronbach"; if you use the abbreviation STM, you may consider also using the SEM.

Discussions - the title should be in singuler and not plural

"On the contrary, the museum's peripheral services of the museum (shopping, dining, guided tour facilities, etc.) do not have a positive impact on the museum visitor personal experience" - try to re-phrase, it is not true that peripheral services do not have a positive impact, rather, they are not that important as the core services. The same goes with the following "ambiance has a negative impact on the visitor personal experience and satisfaction" and "Research has shown that the perpherial services and ambiance of museums tend to have a negative impact on visitors." - not true, only if it is poorly managed. Try to re-phrase.

P. 13, p. 14, p. 15, p. 16, p. 17, p. 18, p. 19 - Error! Reference source not found.

Refer to the tables also in the text.

Reviewer #2: PONE-D-24-02778

Measuring the relationship between museum attributes and visitors from unstructured data: A topic modeling application on museum visitors’ online reviews

Purpose: As on page 9, “this study uses a structural topic modeling approach to identify museum attributes from visitors' online reviews and evaluate the quality of museum services”. The impact of museum attributes on visitor experience and satisfaction was also tested.

Introduction: Why is it that “museums are increasingly threatened by cultural tourism”. Museums are an element of cultural tourism and this statements appears to be counter-intuitive.

It is stated that, “these studies have used structured data for measurement” (page 8). This is not accurate as some previous studies on museums have used Tripadvisor reviews.

Page 9 states, “there are studies that use online reviews to analyse the quality of museum services, most of them use manual analysis, and very few studies use automated analysis to study the quality of museum services”. Why is automated analysis so important? Is this really true?

Literature review: The Literature Review is divided into these parts: visitor experience and museum attributes; visitor satisfaction; visitor experience, museum attributes, and visitor satisfaction: Hypotheses development.

Research design: Qualitative text data on which quantitative methods were applied.

Methodology: Used Structural Topic Modeling and structural equation modeling with data from TripAdvisor on British museums.

Findings: According to the abstract, “while peripheral services and overall ambience have a negative effect on visitor experience and satisfaction. All seven hypothesized relationships were supported in this study, further validating the connection between museum attributes and visitor perception”.

Discussion: This could engage more with the previous research about visitors to museums. For example, there are some sources that find “peripheral services” to be important to visitors, and this study’s findings are contradictory, e.g.,

Li, Z., Shu, S., Shao, J., Booth, E., and Morrison, A. M. (2021). Innovative or not? The effects of consumer perceived value on purchase intentions for the Palace Museum’s cultural and creative products. Sustainability, 13(4), https://www.mdpi.com/2071-1050/13/4/2412#.

Shao, J., Ying, Q., Shu, S., Morrison, A. M., and Booth, E. (2019). Museum tourism 2.0: Experiences and satisfaction with shopping at the National Gallery in London. Sustainability, 11(24), 7108, https://doi.org/10.3390/su11247108

Implications: Two theoretical implications are cited on page 21: 1)the first study to use STM to investigate TripAdvisor reviews in museum, and 2) proposes the impact of museum attributes on visitor personal experience and satisfaction by integrating literature into museum marketing and museum attributes. The use of a specific technique does not necessarily constitute a contribution of a research study; and there are other studies that have considered the influence of museum features on visitor experience and satisfaction. Therefore, the argument on contribution is not strong.

The practical implications are logical based on the findings.

Limitations: This research was based on reviews from Tripadvisor and the authors might have suggested that quantitative research or more in-depth qualitative research should be conducted in the future.

Conclusions: The conclusions are brief; however, if more extended they would probably be repeating passages that appeared before in the manuscript.

Communications (language/style/presentation): “Error! Reference source not Found” appears several times in the manuscript. Overall, the manuscript is rather untidy and should have been more carefully checked prior to submission.

Title and abstract: It is rather a long title and using “unstructured” data seems unnecessary.

Recommendation: Overall, this is a well-implemented research study and could make a good contribution to museum research. However, the manuscript needs to be cleaned up and some of the claims need to be toned down or more fully supported. A major revision seems necessary.

6. PLOS authors have the option to publish the peer review history of their article (what does this mean?). If published, this will include your full peer review and any attached files.

Reviewer #1: No

Reviewer #2: **Yes: **Alastair M. Morrison

---

## [Author Response · Author response to Decision Letter 0]

9 May 2024

Dear reviewers, 

We greatly appreciate your valuable comments and suggestions on our research. Our team has made the correct corresponding changes based on your comments and suggestions and hope that the explanation has fully addressed all of your concerns.

In the remainder of this letter, we discuss each of your comments individually along with our corresponding responses. To facilitate this discussion, we first retype your comments in italic font and then present our responses to the comments. The details are as follows.

Response to Reviewer：

Reviewer #1：

For Comment #1:The recently revised ICOM definition of museums should be provided, the one from 2007 is old. "However, with socio-economic development, museums are increasingly threatened by cultural tourism" - please elaborate on that, it is not clear. The sentence that follows contradicts this "threat".

Our Response #1:Thank you very much for your comments! We have updated the definition of a museum for 2022 (refer to lines 47-50 of the revised manuscript) and outlined the potential threats to museums (refer to lines 51-56 of the revised manuscript).

Recognising the important role of museums in social development and cultural transmission, the International Council of Museums redefined museums in 2022 as 'a non-profit, permanent institution at the service of society that explores, collects, conserves, interprets and exhibits tangible and intangible heritage[4]'. However, with the development of society and the economy, the increase in the number of visitors to object museums has affected the operation of museums, resulting in negative phenomena such as long queues and loud noise, which bring a bad experience to tourists visiting museums[5]. Meanwhile, other cultural tourism products are becoming more interactive and engaging, making the static displays and explanations of traditional museums appear relatively uniform and unappealing[6].

For Comment #2: Hypothesis development - why H2, H3, H5 and H6 would have a negative impact? Yes, but only if they are poorly managed, otherwise they may greatly enhance both the experience and satisfaction. Try to re-phrase the hypotheses. I don't understand why you introduce another hypothesis (H7), besides there is only one hypothesis while you are referring to it in the plural. In general, the paper would benefit from a lower no. of hypotheses, try to reduce and re-phrase.

Our Response #2: Thank you very much for your comments! Your comments is valuable, for H2, H3, H5 and H6, our team has further understood the empirical research paradigm and then we have rephrased the research hypotheses based on relevant literature (Reflected in lines 226-261 in the revised manuscript.). Our explanation as to why we introduced H7 is as follows: personal experience and satisfaction of museum visitors is an interactive and mutually reinforcing relationship. Quality personal experience usually translates into higher satisfaction, and high satisfaction in turn stimulates visitors' desire to revisit and even spontaneous word-of-mouth promotion for the museum. Therefore, when assessing visitor satisfaction with museums, the role of the visitor's personal experience in enhancing visitor satisfaction should be considered.

In addition to the core exhibition, the museum's peripheral services are another important attribute that is an indispensable part of the overall visitor experience. Good peripheral services can eliminate inconveniences during the visit, enhance the visitor's immersive experience, and promote the visitor's sense of identity and belonging to the museum, which can effectively increase visitor satisfaction[12]. In recent years, museum practitioners have been actively engaged in creating a comfortable/healthy external environment, i.e. peripheral services, for their visitors, as this has a direct impact on the visitor experience[47]. In museums, peripheral service factors such as museum location, food and beverage facilities, and ticket reservations respond to visitors in terms of emotion/cognition[48]. Heesup et al also argued that museum peripheral services help visitors to effectively derive intellectual value and satisfactorily evaluate their overall museum experience[49]. As a result, we proposed the following：

H3: The museum peripheral services have a positive impact on the personal experience

H4: The museum peripheral services have a positive impact on visitors satisfaction.

In addition to core offerings and peripheral services, museum ambiance plays an important role in enhancing museum visitation. Museum ambiance is an environment designed to create stimuli that influence the perceptual experience and affective behaviour of visitors[50]. This environment includes factors such as temperature, odor, and lighting, and these environmental conditions are crucial for museum visitors[51]. Wu and Li, argue that these environmental conditions affect visitor experience and satisfaction[52]. Harrison and Shaw analyzed core experience, staff services, peripheral services and facilities, showing that staff services and ambiance had less impact on visitor satisfaction than visitor recommendations to others[53]. However, Huo and Miller concluded that staff service and ambiance had a significant effect on satisfaction[54]. Zanibellato et al. concluded that museum ambiance is an important driver of visitor satisfaction, while museum layout, temperature, and queuing are the key factors that influence the visitor's experience of visiting the museum[12]. According to Nurfatihah et al. the ambiance of a restaurant provides a pleasant experience for visitors[55]. It is believed that ambiance has an important mediating role in museum visitors and can enhance the attractiveness of museum[56]. In this regard, the following hypothesis were proposed in this study :

H5: The museum ambiance have a positive impact on the personal experience.

H6: The museum ambiance have a positive effect on visitors satisfaction.

For Comment #3: Some language editing is required (e.g. spaces between words and brackets; "Kempiak et al., (2017)shows" - show; "it to others[28-35]. whether" - capital W; "Hypothese" - hypothesis; "Lin, (2009) the study identifies" - ???; "core offerings has" - have; "peripheral services has" - have; "all belong to one of the top ten most visited museums" - do you mean "all belong to the top ten most visited museums"; "perpherial" - peripheral; etc.).

Our Response #3: Thank you for your critical review and valuable suggestions regarding the linguistic presentation of my manuscript. I apologise for the many details that you have pointed out, which do reflect an oversight in the standardisation of the language of the manuscript, and we have revised them. Due to the number of revisions, it is not convenient to go into detail in the Q&A section, which can be viewed in a revised manuscript.

For Comment #4: This is not a sentence: "Determine the appropriate number of factors (museum attribute categories), according to Kaiser, (1960) eigenvalue rules (keeping factors with eigenvalues greater than 1)[60]." - add the missing parts.

Our Response #4: Thank you very much for your comments! Based on your valuable suggestions, we have revised the sentence to convey the message more accurately and completely.(Reflected in lines 369-377 in the revised manuscript.)

During the extraction phase, principal component analysis was performed using orthogonal rotation. The eigenvalue criterion was used in this study to identify the relevant dimensions of museum attributes, this criterion recommends retaining only communal factors with eigenvalues greater than one[71]. In factor analysis, each underlying factor is associated with an eigenvalue derived from the covariance matrix, if the eigenvalue of a factor exceeds one, it indicates that the factor explains more variability than any individual raw variable, thus justifying its inclusion in the final factorial model[71].

For Comment #5: Methodology - "Cranbach" - it is "Cronbach"; if you use the abbreviation STM, you may consider also using the SEM.

Our Response #5: Thank you very much for your comments! We apologise for these writing irregularities, and have revised the manuscripts.

The first step involves checking the Cronbach coefficients for individual items, which should all exceed a threshold of 0.7[74].(Reflected in lines 406-408 in the revised manuscript.)

This study utilizes SEM to test the research model, ensure model fit, and test the proposed hypothesis. SEM is valid for path analysis models with moderating and mediating variables[65].(Reflected in lines 347-349 in the revised manuscript.)

For Comment #6: Discussions - the title should be in singuler and not plural.

Our Response #6: Thank you very much for your comments! We apologise for not having carefully reviewed the contents of the manuscript. We have provided a more accurate revision of the manuscript on this issue.(Reflected in line 437 in the revised manuscript.)

For Comment #7: "On the contrary, the museum's peripheral services of the museum (shopping, dining, guided tour facilities, etc.) do not have a positive impact on the museum visitor personal experience" - try to re-phrase, it is not true that peripheral services do not have a positive impact, rather, they are not that important as the core services. The same goes with the following "ambiance has a negative impact on the visitor personal experience and satisfaction" and "Research has shown that the perpherial services and ambiance of museums tend to have a negative impact on visitors." - not true, only if it is poorly managed. Try to re-phrase. 

Our Response #7: Thank you very much for your comments! We have made the appropriate changes as you suggested. The specific modifications are as follows: (Reflected in lines 464-466、486-497 in the revised manuscript.)

Museum peripheral services (shopping, dining, guided tour facilities, etc.) have less of a positive impact on the personal experience and satisfaction of museum visitors compared to core offerings.

Zanibellato et al. identified the sub-attributes of ambiance, such as crowding, queuing, and visiting time, as key factors that affect visitor satisfaction[12]. Shao, et al. pointed out in his study that the factor of crowding can have a significant negative impact on visitors' experience and satisfaction[78]. The sub-attributes below the ambiance are mainly caused by the museum's mismanagement. For example, the failure to control the number of visitors results in long queues and reduces the time available for tourists to visit the museum. Additionally, the lack of reasonable paths in the browsing area leads to overcrowding and other issues. Mismanagement can have a negative impact on visitors' personal experience and satisfaction. Therefore, museum managers should take measures to create a favourable museum ambiance for visitors to enhance their experience and satisfaction.

For Comment #8: P. 13, p. 14, p. 15, p. 16, p. 17, p. 18, p. 19 - Error! Reference source not found. Refer to the tables also in the text.

Our Response #8: Thank you very much for your comments! I'm very sorry I didn't notice this before, but the references have now been updated throughout the text. It can be viewed in a revised manuscript. Due to the number of revisions, it is not convenient to go into detail in the Q&A section, which can be viewed in a revised manuscript

We would like to take this opportunity to thank you for the time you have invested in our article and for giving us the opportunity to improve the manuscript. We hope you will find this revised version satisfactory. If you have other improvement ideas for our revised manuscripts, please let us know in time. We are willing to make further revisions to improve the quality of this paper. 

Thanks again for your help. 

Best regards, 

All authors

Reviewer #2：

For Comment #1:As on page 9, “this study uses a structural topic modeling approach to identify museum attributes from visitors' online reviews and evaluate the quality of museum services”. The impact of museum attributes on visitor experience and satisfaction was also tested.

Our Response #1: Thank you very much for your comments! Due to a writing error, we have now added the test of visitor personal experience on satisfaction to the manuscript. (Refer to lines 98-102 of the revised manuscript).

Then, the manuscript categorised the extracted topics using existing literature and factor analysis. The impact of museum attributes on visitors' personal experience and satisfaction, as well as the impact of visitors' personal experience on satisfaction, were analysed using exploratory factor analysis (EFA) and structural equation modelling.

For Comment #2: Why is it that “museums are increasingly threatened by cultural tourism”. Museums are an element of cultural tourism and this statements appears to be counter-intuitive.

Our Response #2: Thank you very much for your comments! We made the following explanations, first of all, we explained that why "museums are increasingly threatened by cultural tourism", which is mainly because museums are a kind of cultural tourism, and other cultural tourism keeps on innovating and developing more and more attractive tourism products for tourists, which reduces the attractiveness of museums.(Reflected in lines 54-56 in the revised manuscript). The definition of museums as a type of cultural tourism is based on previous studies.(Refer to lines 44-45 of the revised manuscript）

Meanwhile, other cultural tourism products are becoming more interactive and engaging, making the static displays and explanations of traditional museums appear relatively uniform and unappealing[6]. 

Museum tourism is a crucial component of cultural tourism and provides visitors with an opportunity to learn about the cultural and historical aspects of a city[1].

For Comment #3: It is stated that, “these studies have used structured data for measurement” (page 8). This is not accurate as some previous studies on museums have used Tripadvisor reviews.

Our Response #3: Thank you very much for your comments! We apologise that we did not carefully review the references cited. We have provided a more accurate reference for this issue. (Reflected in lines 62-65 in the revised manuscript.)

Currently, there are many scholars who have studied the experience and satisfaction of museum visitors through questionnaires and interviews, as well as to understand visitors' motivations and intentions towards the museum[7-8]. These studies use numerical data.

For Comment #4: Page 9 states, “there are studies that use online reviews to analyse the quality of museum services, most of them use manual analysis, and very few studies use automated analysis to study the quality of museum services”. Why is automated analysis so important? Is this really true?

Our Response #4: Thank you very much for your comments! There are two primary reasons for selecting automatic text analysis in this study. Firstly, the sample size of data is large, and automatic analysis can enhance the efficiency of data processing. Secondly, automatic text analysis follows a unified standard algorithm, which reduces the subjective differences introduced by manual analysis and ensures the consistency and accuracy of the analysis results. Reference 11 compares the advantages and disadvantages of the two analysis methods.(Reflected in lines 71-76 in the revised manuscript)

In the context of museum research, studies analysing the quality of museum services using online reviews are predominantly manual, with very few utilising automated analysis. Due to the large volume of online comments, manual analysis can be time-consuming and labour-intensive, and it may introduce subjective evaluations, reducing the accuracy of the text analysis, automated analysis is more objective and reliable, and can save time[11].

De Graaf R, van der Vossen R. Bits versus brains in content analysis. Comparing the advantages and disadvantages of manual and automated methods for content analysis. Communications. 2013;38(4):433-43.

For Comment #5: Discussion: This could engage more with the previous research about visitors to museums. For example, there are some sources that find “peripheral services” to be importan

---

## [Decision Letter · Decision Letter 1]

21 May 2024

Measuring the Relationship between Museum Attributes and Visitors: An a pplication of Topic Model on Museum Online Reviews

PONE-D-24-02778R1

Dear Dr. Shen,

We’re pleased to inform you that your manuscript has been judged scientifically suitable for publication and will be formally accepted for publication once it meets all outstanding technical requirements.

Kind regards,

Andrea Fronzetti Colladon, Ph.D.

Academic Editor

PLOS ONE

Reviewers' comments:

Reviewer's Responses to Questions

**Comments to the Author**

1. If the authors have adequately addressed your comments raised in a previous round of review and you feel that this manuscript is now acceptable for publication, you may indicate that here to bypass the “Comments to the Author” section, enter your conflict of interest statement in the “Confidential to Editor” section, and submit your "Accept" recommendation.

Reviewer #1: All comments have been addressed

Reviewer #2: All comments have been addressed

2. Is the manuscript technically sound, and do the data support the conclusions?

Reviewer #1: (No Response)

Reviewer #2: Yes

3. Has the statistical analysis been performed appropriately and rigorously? 

Reviewer #1: (No Response)

Reviewer #2: Yes

4. Have the authors made all data underlying the findings in their manuscript fully available?

Reviewer #1: (No Response)

Reviewer #2: Yes

5. Is the manuscript presented in an intelligible fashion and written in standard English?

Reviewer #1: (No Response)

Reviewer #2: Yes

6. Review Comments to the Author

Reviewer #1: Minor language editing is still required (e.g. spaces missing between sentences, or sentences enging with a comma instead of a full stop).

Reviewer #2: Thank you for addressing my comments so comprehensively. I have read your responses and I believe they address the concerns that I had with the original manuscript.

7. PLOS authors have the option to publish the peer review history of their article (what does this mean?). If published, this will include your full peer review and any attached files.

Reviewer #1: No

Reviewer #2: **Yes: **Alastair M. Morrison
